# Understanding How Marine Protected Areas Influence Local Prosperity—A Case Study of Gili Matra, Indonesia

**DOI:** 10.3390/ijerph192013508

**Published:** 2022-10-19

**Authors:** Amrullah Rosadi, Paul Dargusch, Taryono Taryono

**Affiliations:** 1School of Earth and Environmental Science, The University of Queensland, Brisbane, QLD 4072, Australia; 2Indonesia Climate Change Trust Fund (ICCTF), Ministry of National Development Planning/Bappenas Republic of Indonesia, Jakarta 12940, Indonesia; 3Department of Aquatic Resources Management, Faculty of Fisheries and Marine Sciences, IPB University, Bogor 16680, West Java, Indonesia

**Keywords:** marine protected area, prosperity, marine tourism, stakeholder, management approach, mental model

## Abstract

A Marine Protected Area (MPA) is always expected to create a specified outcome in ecosystem improvement. While they are certain to benefit marine life, MPAs also impact the surrounding communities, as they directly affect the livelihoods of people who rely on marine exploits to make a decent living. In other words, MPAs create new communal dynamics influencing the rate of prosperity in the surrounding communities. Unfortunately, the leverage of MPAs in the coastal communities’ social economy is often under-assessed in MPA-related research. The MPA’s influence on communal prosperity emphasizes the importance of policy incentives from stakeholders. Therefore, stakeholders’ perceptions of MPAs are fundamental in the planning and implementation of MPAs, which could improve the prosperity of the coastal communities. In Gili Matra, Indonesia, where tourism is the MPA’s backbone, MPAs are expected to sustain prosperity for future generations. However, some stakeholders with different influential stances to the MPA (Influential Stakeholders (IS) and Non-Influential Stakeholders (NIS)) demand a contradictive approach. This could lead to managerial issues for the MPAs. These issues must be addressed to avoid contradictory objectives that could harm MPA implementation.

## 1. Introduction

Through the protected area approach, conservation programs influence the dynamic of coastal communities’ social and economic situations, thus affecting people’s prosperity. Human aspects such as social, economic, and cultural values are fundamental for protected area management [1]. The tourism-based protected area is one example where conservation programs fluctuate the community’s prosperity, as benefits and issues may emerge alternately. One of the critical issues regarding tourism-based protected areas is its economic benefit, as the uncontrolled number of visitors threatens the socio-cultural context [2]. Through qualitative assessment, ecotourism in protected areas stimulates infrastructure revitalization, such as building schools and community health centers, which are crucial to improving communities’ overall prosperity [3].

Taking a broader example, the long history of environmental conservation and the protected area program in Nepal had influenced prosperity on a national and local scale [4]. The protected areas’ revenue could build community and infrastructure development programs locally by providing road construction, education, training and capacity building, health and sanitation, livelihood, and water access [4]. Meanwhile, on a national scale, the effective management of protected areas influenced the country’s prosperity through GDP and employment rate increases [4].

A similar approach to conservation objectives is applied to Marine Protected Areas (MPAs). It targets ecosystem benefits such as coral reef health improvement or sea turtle population increases and expects local communal prosperity to flourish. Besides aiming for ecosystem health in the ocean or preserving vulnerable marine species, MPAs are also designed to address social and economic goals and objectives [5]. The collaborative effort of an MPA, which, in this case, aimed to reduce overfishing, would benefit fishers in the long term [6]. By strengthening community rights for coastal development, the conservation approach for MPAs is the bridge between human welfare and the environment [7].

Through a study in the Bunaken MPA, it was proved that MPAs could contribute to prosperity improvement by alleviating poverty [8]. Three poverty domains (security, opportunity, and empowerment) [9] were improved in the period after MPA management was introduced to the region. MPA management successfully improved the coastal community’s daily earnings in the security domain [8]. Comparatively, in the opportunity domain, the coastal communities within the MPA region achieved a slight rise in wealth, environmental knowledge, and fishing [8]. Regarding empowerment, the coastal communities were well equipped to perform natural resource control and influence community affairs concerning MPA management [8].

The crucial MPA aspect of prosperity improvement in the communities is the MPA’s policy’s capital value. MPAs could contribute to poverty alleviation in coastal areas through policy incentives [10]. The policies should be utilized for infrastructure investment within the MPA area, coastal communities’ empowerment, collaboration, and networking, among other MPAs [10].

As the policy incentives are the key to prosperity resulting from MPA management, the MPA stakeholders’ understanding of how MPAs influence coastal communities’ welfare is fundamental to ensuring the MPAs’ socio-economic objectives. Generally, stakeholders’ views on prosperity resulting from MPA emphasize the prioritization of MPAs to achieve ecosystem health objectives; thus, prosperity in the local community can emerge following the success of ecological gains [11,12].

However, the social and economic dimensions are often neglected or examined less systematically [13]. Only a limited number of studies assess the effect of MPAs on prosperity-related topics, while most scientific studies on MPAs focus on the biological effects on marine conservation reserves [14].

Therefore, before advancing the MPA works to increase social and economic benefits for the local communities, it is fundamental to first reveal the process of MPA influences concerning communal prosperity by acknowledging the key stakeholders as policy incentive managers in the MPA. This could be carried out by exploring the following questions: (1) What key stakeholder groups influence and are influenced by the MPA? (2) What is currently known about how the MPAs influence local prosperity? (3) How do key stakeholders understand how the MPAs influence local prosperity?

According to the Ministry of Marine Affairs and Fisheries (MMAF), Gili Matra Marine Tourism Park (hereafter, “Gili Matra MPA”) has been one of the Indonesian MPAs since 2009 under the marine tourism park management approach, covering 2954 ha of ocean boundaries [15]. One of the Gili Matra MPA’s conservation targets is to preserve the sustainability of the essential marine ecosystem, such as coral reefs, mangroves, seagrasses, and fishery resources. The Gili Matra MPA’s conservation target also includes local communities, fishers, and tourism operators [15]. The Gili Matra MPA has significant ecotourism activity supported by modern facilities [16].

Considering the surge of ecotourism activities in Gili Matra’s MPA, this research hypothesizes that the MPA management could provide social and economic dynamics within the coastal communities and influence their prosperity. Therefore, the Gili Matra MPA is compatible with the abovementioned questions as a case study. This research will examine the perspectives of the stakeholders of the Gili Matra MPA regarding the process of improving local prosperity in their surrounding communities through the Gili Matra MPA management.

## 2. Materials and Methods

This research focuses on a qualitative assessment addressing the research aims and objectives, which requires primary data from stakeholders’ in-depth interviews. The interview processes are audio-taped, transcribed, and translated into English.

As this research examined the stakeholders of the Gili Matra MPA, the primary data collection took place within the area of the MPA and its surrounding areas. The Gili Matra MPA is comprised of three islands of the North Lombok District, Indonesia, namely, Gili Trawangan, Gili Meno, and Gili Air, within the boundary of 116°01′34″ E to 116°05′18″ E and 8°20′02″ S to 8°22′16″ S. Figure 1 presents an MPA zonation map of the Gili Matra MPA, as provided by the MMAF [15].

### 2.1. Stakeholder Identification and Mapping

Stakeholder identification is an essential part of learning the process of change initiative [17]. Stakeholder mapping is a fundamental tool for assessing the policy intervention process [18]. Stakeholder identification and mapping are beneficial for conservation project assessment [19,20]. Therefore, this research acknowledges stakeholder identification and mapping as critical tools in examining the stakeholder perspectives on MPA benefits by using the Cross-Cutting Tool Stakeholder Analysis and Influence-Interest grid [21,22] as the initial sequence of the research. 

The mentioned methods [21,22] use influence and interest axes and are divided into four categories of stakeholders. The first category of stakeholder is the Key Player, comprising those with high influence and high interest. Stakeholders with less interest but strong power fall under the second category: Context Setters. The third comprises stakeholders with low influence and high power, called Subjects, while the remaining category contains stakeholders with low power and interest. Figure 2 shows the grid that is used for the stakeholder mapping of this study:

The study plots the stakeholder position into the grid by providing a score from 1 to 5 for their influence and interest based on the following adapted indicators for stakeholders’ relationships with conservation management (Table 1).

The total scores for each stakeholder (1–20) will indicate their position towards the Gili Matra MPA management based on their position in the grid (Figure 2). 

In this research, eighteen stakeholders were identified and mapped with the following criteria: (1) the stakeholders whose primary responsibilities are to authorize and manage the MPA approach, including ensuring the policy incentives; (2) the stakeholders with legal jurisdiction and administrative responsibilities in human and natural resource management; (3) stakeholders interested in the Gili Matra MPA; and (4) the stakeholders representing the coastal communities that aim to establish advocacy for natural resource management within the MPA boundaries. 

### 2.2. Current Conditions and Mental Model

The qualitative assessment in this research was implemented through in-depth interviews which were conducted in March 2022. Eighteen interviews were conducted with eighteen stakeholders and were used as the primary sources to identify the stakeholder perspectives on the MPA’s influences on prosperity in Gili Matra. The mentioned interviews were designed as open-ended and semi-structured oral interviews, where the participating stakeholders were asked numerous probing questions [25] about the MPA’s influence on local prosperity in Gili Matra. The data collected from the interviews were sorted using a data sources triangulation approach to find the perspectives on the circumstances, process, and integration from each stakeholder elicitation [26]. Finally, the result of the data collection is illustrated through the mental model.

The mental model represents the reality process from the external in a cognitive way [27]. They also observed that, in the natural resource management domain, mental model elicitation could accommodate the plurality linkages between the values and goals for a range of stakeholder perceptions in the mechanism of the resource management system. In the conservation and protected area approach, the mental model is fundamental to revealing how the beneficiaries understand the system, including its content and its structure [28]. The mental model also supports protected area governance and management by conceptualizing information flow in the regulation enhancement process [29].

As such, a mental model will be used to examine the perspectives of selected stakeholders concerning the process and mechanism of policy incentives from MPA management in Gili Matra, analyzing how MPAs can help improve prosperity.

This report will explore the mental model based on three domains of prosperity from the World Bank [9], which are opportunity, empowerment, and security. The opportunity domain is explored by identifying the income, education, and fish caught in the MPA area; the security domain is comprised of community union and health and their resiliency to crisis; and the empowerment area contains public participation in the institutional process and capacity building [8].

## 3. Results

The research results are limited to the mental model on the MPA’s influence over prosperity, which is shaped by the perception of each stakeholder category. The stakeholder categories are arranged by the output from the stakeholder mapping in the Gili Matra MPA and the general findings of the mentioned mental model. The stakeholders in the Gili Matra MPA were divided into four groups based on their scores on the influence and interest in the MPA, as shown in Table 2 and Figure 3.

### 3.1. Stakeholder Mental Model on the MPA Influencing Local Prosperity

In general, the mental model regarding the MPA’s influence on local prosperity in Gili Matra could be divided based on the fundamental differences between the stakeholder groups outlined above. The mental model result provides a significant difference in perspective regarding the influence of the MPA on prosperity between the Key Player–Context Setter and Subjects–Crowd. On the other hand, there is no perspective difference between the Key Player and Context Setter, nor is there a difference between the Subjects and Crowd stakeholders. Therefore, considering that the stakeholder mapping result identifies distinct differences between them are the influence score in Gili Matra, this report will reveal, discuss, and classify the mental model of the influential stakeholder (Groups 1 and 2) and non-influential stakeholder groups (Groups 3 and 4). 

#### 3.1.1. Opportunity Mental Model of Influential Stakeholders

Primarily, the influential stakeholders (IS) think that the MPA is one fundamental way to sustain Gili Matra’s existing prosperity. Prosperity regarding opportunity is believed to be achieved because of the massive marine tourism in Gili Matra. Tourism is the primary factor of economic activity in Gili Matra and significantly contributes to the community’s daily earnings:

Gili Matra has been well known for its mass tourism since around 1990; the tourism has caused the community to live much more prosperously. This is the main source of prosperity in the area.(IS 1, 2022)

Gili Matra is one of the world-class tourism destinations. Its potency has led to it being included as one of the national strategic tourism areas. This makes Gili Matra one of the largest contributors for district revenue.(IS 3, 2022)

The prosperity in the MPA could be achieved on the basis of the livelihood formed from the marine tourism; there is a wide range of jobs and economic activity derived from the tourism in the form of guides, accommodation, catering, attraction, and transport.(IS 2, 2022)

The accumulation of these circumstances could improve the communities within Gili Matra by providing higher education to their descendants. With the rise in income, the tourist actors in Gili Matra can provide higher education for their children compared to past generations. However, this education improvement was only achieved by sending their children to the mainland of Lombok, as Gili Matra is limited in its school infrastructure. The MPA establishment still faces limitations regarding this issue:

The responsibility for establishing proper infrastructure for education belongs to the district government, not the MPA.(IS 2, 2022)

Even until recently, the education infrastructure in Gili Matra has been limited. However, the money from tourism enables them to send their kids to better schools in the mainland; some are even able to pay for education abroad.(IS 3, 2022)

Unfortunately, the massive tourism in Gili Matra comes with potential issues in regard to the education of local communities. Providing tourism activities in Gili Matra creates a comfort zone for youth. Some teenagers in Gili Matra MPA consider working in tourism as a better pursuit than earning a degree in higher education. Some even stop after junior high school and refuse to advance to the secondary education level. Specific stakeholders believe MPA management could do more to improve the attitude of these students.

However, although the massive tourism in Gili Matra enables the people to afford an education, it is worrying to see that some of them are not willing to continue their school due to their ability to earn money in the tourism business without a school degree. This issue may be unrelated to the MPA, but it will be much better if the MPA can regulate something to solve this issue.(IS 4, 2022)

#### 3.1.2. Security Mental Model of Influential Stakeholders

The influential stakeholders believe that the tourism activity in Gili Matra creates a resilient community that unites against crises, which is a security factor of prosperity. Long years of managing tourism have shaped the livelihoods of Gili Matra’s people, and as such, they quickly recover from crises. This is reflected by the 2018 earthquake in Gili Matra, where tourism was able to recommence almost immediately after the disaster occurred. Besides this resiliency, the security dimension is also affected by the strong sense of social organization. This is due to the MPA’s ability to provide empowerment to the people. 

Involving the community is essential to ensuring the effectiveness of MPA management. Allowing the community to participate in the management approach could be carried out through encouraging the establishment of thematic community groups, such as the sustainable tourism group, aquaculture group, fisher group, and fish processing group, and through providing them channels to government grants. The mentioned scheme could encourage people to strengthen their organizing skills for social life.(IS 1, 2022)

In some circumstances, the MPA can potentially channel the grant facility from the national government to the people. Unfortunately, the facility cannot be processed for the individual. Social organization is necessary to distribute this facility, and the community is encouraged to establish a group to receive this facility. These circumstances are considered the stage where prosperity in the security domain is reached, as the community is strengthened.

The crisis resiliency of the Gili Matra people could be high, but it depends on the type of crisis. Reflecting on the massive earthquake in 2018, the people in Gili Matra only needed two months to recover and had the tourism activities back to normal. I conclude that the years of experience in managing tourism have caused the people in Gili Matra to adapt, survive, and work together in recovering Gili Matra whenever crises arise.(IS 3, 2022)

The massive tourism activity in MPA triggers the private sector to invest in health infrastructure in Gili Matra. The MPA may also have contributed to some public health facilities, but it is not significant.(IS 2, 2022)

Since the tourism in Gili Matra proliferated, the private investor’s health facility has grown. Some health facilities from the state are also available. Additionally, Gili Matra also has better sanitation by installing wastewater treatment facilities in multiple sites within the area.(IS 4, 2022)

While the tourism industry has its economic benefits for the Gili Matra people, uncontrolled tourism may create substantial environmental degradation. As a result, Gili Matra may no longer be appealing to tourists, which will devastate the community’s income and affect the livelihoods of its citizens. This is why the influential stakeholders believe that the MPA is fundamental in sustaining Gili Matra’s ecosystem. So long as it remains healthy and stimulates the tourism industry in Gili Matra, it will influence the prosperity of its people.

I am seeing that the MPA in Gili Matra is a government approach to preserving the coral reefs and other marine ecosystems through controlling the utilization of the natural resources.(IS 4, 2022)

The conservation approach in Gili Matra is essential to sustaining tourism. We believe that, without proper control, the ecosystem in Gili Matra will be damaged and no longer support tourism. The people will therefore suffer because they lost their job and their prosperity plummeted.(IS 3, 2022)

From the tourism activities in Gili Matra, numerous economic activities occur from the accommodation, transportation, catering, and guiding, which create livelihood variability. This is the point where the prosperity of the people could arise. However, people have to realize that the actual attraction of Gili Matra is the beauty of the marine ecosystems. Therefore, it is fundamental to preserve the ecosystem to sustain tourism so that the people can prosper.(IS 5, 2022)

The main purpose of the MPA is to create the effectiveness of marine resource management, not only for the environment but also for the surrounding people; thus, the communities could receive benefits sustainably, which includes living prosperously.(IS 1, 2022)

Additionally, the MPA also believes in creating alternative livelihoods through fishing. It understands that the people of Gili Matra were originally fishers before relying on tourism. Providing an alternative livelihood through fisheries will allow for security whenever tourism is low. The MPA establishment is thought to be the way for fish stocks in nature to increase. The stakeholders believe that as the people’s education and fish stock increase, alternative ways of living will emerge and provide security for the people to live prosperously.

Considering that one of the MPA’s aims is to enrich the population of the fish in the ocean, I believe that, in achieving MPA conservation objectives, the prosperity of the communities in Gili Matra will be automatically increased through alternative livelihoods in fisheries.(IS 5, 2022)

#### 3.1.3. Empowerment Mental Model of Influential Stakeholders

In the domain of empowerment, the prosperity of Gili Matra is indicated through the MPA mechanism, where community participation is required to design the MPA. A wide range of MPA policies requires consultation with the people before its endorsement. The mandatory consultation process will train and encourage the people in Gili Matra to understand the political and institutional bureaucracy:

Fundamentally, the MPA design process contains public consultation as an important mechanism to finalize the zonation system. Therefore, the MPA will automatically support the public participation in the institutional process.(IS 1, 2022)

Establishing the MPA in Gili Matra is included as a central government approach; therefore, a wide range of community empowerment funding programs should be available.(IS 4, 2022)

The experience and knowledge of the communities about the touristy spots in the area are beneficial to improving the MPA’s tourism. Therefore, by encouraging community groups, the MPA in Gili Matra is obligated to retrieve and consider the people’s data and information for MPA improvement.(IS 2, 2022)

Figure 4 illustrates the flow of influential stakeholder perspectives regarding the process of MPAs influencing prosperity in the form of a mental model.

### 3.2. Mental Model of Non-Influential Stakeholders

Fundamentally, the non-influential stakeholders (NIS) have a similar basic perception of the MPA’s influence on the prosperity in Gili Matra, where the MPA’s role is to sustain the tourism activity and stabilize the community’s income. The communities acknowledge that the conservation approach is vital for the future of the communities’ prosperity in Gili Matra, and it could be implemented through the MPA.

The aim of the MPA itself is to create sustainability management of the ocean ecosystem and enable the communities to derive income from it without damaging the marine environment through a zoning system.(NIS 1, 2022)

This MPA in Gili Matra is a government regulation approach to ensure sustainable practice for the utilization of the marine resources by communities; thus, the ecosystem services will remain positive in influencing the community’s social economy in the future.(NIS 2, 2022)

#### 3.2.1. Opportunity Mental Model of Non-Influential Stakeholders

In the opportunity domain, the prosperity in Gili Matra was initially reached through the economic activity from mass tourism, which also created a domino effect on the affordability of education and health services. The situation also indirectly affects the security domain of prosperity by improving crisis resiliency. Similarly, the community of Gili Matra believes that the MPA could be the sustaining mechanism in preserving the current prosperity for future generations in Gili Matra.

The people in Gili Matra already prospered enough due to their income from the tourism activity, even before the MPA was established. However, uncontrolled tourism also becomes a concern, as it potentially damages natural beauty, especially the coral reefs ecosystem in Gili Matra, and threatens people’s livelihoods. Therefore, we believe that whenever the MPA objectives are achieved, the prosperity of the people will be stable in the future.(NIS 6, 2022)

The education infrastructure in Gili Matra is enough, and the MPA is not responsible for adding more facilities. Therefore, the younger community should pursue higher education on the mainland, and their parents can afford the money from the tourism here.(NIS 4, 2022)

The most directed MPA impact in Gili Matra regarding the education sector is non-formal education regarding marine conservation knowledge. The MPA authority could contribute to improving the capacity of Gili Matra youth’s through the special session in school and spread the essential information about the sustainability of the ocean’s ecosystem.(NIS 11, 2022)

#### 3.2.2. Security Mental Model of Non-Influential Stakeholders

In the security domain, the stakeholders with low influence think that the community’s response to the crisis is well shaped due to its many years of experience in managing tourism and its survival instincts.

The crisis resiliency of the Gili Matra people can be seen through the earthquake in 2018. Their capacity to manage tourism and their sense of survival united them to recover Gili Matra. Therefore, the tourism activity was restored shortly after the disaster.(NIS 1, 2022)

They also acknowledge that a healthy environment affects the community’s overall health. Private health facilities are expanding due to mass tourism. However, the MPA could improve on this by contributing to public health infrastructure in Gili Matra.

The health clinic has been growing here since tourism in Gili Matra became very popular, but it is limited to the private clinic, while public health infrastructure is still limited. Considering the MPA status, public health facilities should be more accessible here.(NIS 12, 2022)

If the objective of the MPA in improving the quality of the environment is achieved, the health quality of the people automatically increases. A clean environment is the source of a healthy body.(NIS 2, 2022)

The experience of having tourism plummet during the COVID-19 pandemic shaped the stakeholders’ thoughts on fisheries as their alternative livelihood. They realize that fishing is their origin of livelihood, and during the tourism crisis, they could temporarily turn themselves back to fishers to provide daily necessities until tourism recovers. This situation somehow makes them more aware that the limitation in MPA zonation will provide more abundant fish and a more secure future in Gili Matra.

The MPA approach contains several no-take zones that enrich the fish population in Gili Matra’s waters. It is expected that the fish could be one of the ecosystem services to sustain the income of the people in Gili Matra.(NIS 3, 2022)

Gili Matra is blessed with outstanding nature and ecosystem services. During the COVID-19 crisis, we went back to the sea to capture fish to survive. Therefore, conserving the ocean through the MPA is essential for us to live in the future.(NIS 8, 2022)

#### 3.2.3. Empowerment Mental Model of Non-Influential Stakeholders

The NISs acknowledge that community participation in the political and institutional process increased after the MPA was established. The stakeholders believe that the empowerment domain of prosperity is affected as a result.

The MPA mechanism to socialize and spread the essential information of the conservation approach implementation created the opportunity for people here to voice their aspirations. The community also provided their opinions about the zonation design on the MPA. As far as we understand, the mentioned process is called public consultation to improve public participation.(NIS 5, 2022)

Aside from the consultation phase of the zonation system, the MPA also requires the people to establish community groups to receive the training and government grant packages for community development. The circumstances empower the people to unite and collaborate in a semi-formal organizational scheme.(NIS 10, 2022)

However, the difference between the influential stakeholders and the NISs is that the latter believe that the MPA delivery should be through a bottom-up approach or managed by local human resources. Some emerging perspectives emphasized that the current top-down approach does not reflect the community’s prosperity goals. From their perspective, the top-down approach could neglect local community participation in the MPA management and potentially release authoritative policy. Although the current MPA management provides multiple consultations for MPA policy and regulations, there are some cases where communities are not well involved in the policy formulation. Consequently, the stakeholders believe that the best way for the Gili Matra MPA to positively influence prosperity is by establishing a locally managed MPA. This would mean the community designs all the protection regulations and that the cultural practices are enforced and managed by them.

However, there are unmatched perspectives between the community practice and the regulation. For example, the community acknowledges the MPA and zonation system, but at the same time, they draw back from the MPA entrance fee regulation, which potentially decreases the tourism enthusiasm in Gili Matra.(NIS 2, 2022)

Although the MPA zonation design is consulted with the public, there are some other regulations that were not approved by the people, such as the MPA entrance fee regulation.(NIS 9, 2022)

In some other areas, the local tradition could strengthen the MPA implementation. Law enforcement and control mechanism value from customary and indigenous knowledge should benefit the MPA to achieve a greater impact and gain people’s trust. Unfortunately, the MPA in Gili Matra does not really apply the value.(NIS 11, 2022)

We understand that the marine ecosystem should be preserved and conserved; thus, the tourism circumstances would remain for decades. However, so far, we have only seen restrictions and higher entrance fees. It makes us wonder whether their objectives are to sustain the ecosystem and the coastal communities’ lives or whether they only want to increase the government income. It seems that the regulation on entrance fees is standardized for all marine national parks in Indonesia, while each marine national park has different characteristics. The current entrance fee regulation cannot be applied in Gili Matra as the low–middle traveler’s tourist characteristic. It is hard to see the direct impact of the entrance fee for Gili Matra. Therefore, instead of following government regulations, we should develop our own initiative to protect Gili Matra’s ecosystem.(NIS 13, 2022)

My concern is when people from the outside, especially the central government, come and try to manage the marine ecosystem here; it does not seem to work. The political situation often creates human resource turnover for the MPA person in charge of Gili Matra. Those circumstances would not achieve strong relationships. Their presence in Gili Matra is still ineffective; we rarely see them here. Therefore, local people in Gili Matra should be more involved in the long term to manage the MPA in Gili Matra instead, as they will provide more presence and control. We also have cultural practices to manage the ocean that are also applicable to MPA implementation.(NIS 7, 2022)

In summary, the perceptions of the NIS are illustrated in the mental model, as shown in Figure 5.

## 4. Discussion

With powerful influence and interest, the first group of stakeholders in the Gili Matra MPA are comprised of the primary MPA authority. They are representative of the national government of Indonesia, which receives the mandate to advocate for the establishment of an MPA and MPA management to achieve its objectives of ecosystem conservation. The tourism agency is also categorized in this group, as they are responsible for releasing licenses and permissions for tourism activity. The tourism agency’s role also functions to maintain the comfort of Gili Matra as an international tourism destination. Therefore, both stakeholders understand Gili Matra’s circumstances enough to manage and implement conservation programs and economic activities within the area. Stakeholders in this group, as the “Players”, require ultimate attention for a practical management approach [21]. The actors in this type of stakeholder group typically belong to the multi-level government sector, which drives the whole process of the collaborative development approach [30].

On the other hand, the second group represents the local government agency that works indirectly to support the MPA’s performance in Gili Matra. The stakeholders in this group have authority; in particular, they have governmental management in Gili Matra, with less interest in conservation. As a government agency, their work is relatively standard, but their released policy for the development potentially affects the Gili Matra MPA’s circumstances. As an illustration, the Environmental Agency is responsible for collecting and managing domestic waste in the North Lombok District, including Gili Matra. Waste collection and disposal is a regular task for any municipal council, but if they decide to no longer support the waste management in Gili Matra, it will impact a wide range of MPA management aspects and create a contradiction to MPA objectives. This type of stakeholder group, which has a strong influence but low power, is considered a context-setter, potentially transforming into the Players if given more awareness and authority [31].

The third stakeholder group, comprising those who have high interest but low influence [21], is comprised of conservation activists, local NGOs, and academicians. The organizations in this group may not have enough power to direct the MPA management, but most are dedicated institutions exploring the value of the Gili Matra MPA from a wide range of perspectives, including ecosystem preservation and restoration. Therefore, their work in Gili Matra is essential to improving the Gili Matra MPA’s capacity to achieve its conservation and socio-economic goals. Besides NGO and conservation activism, the Village Officials were also included in this stakeholder group. The Village Officials are responsible for managing the administrative necessities for Gili Matra’s people. Their duties are less likely to affect the MPA management, but their interest in developing the area’s community supports the MPA works indirectly. As a subject of a collaborative approach to conservation work, this stakeholder group will provide favorable circumstances for cooperation with a wide range of transfer knowledge [32].

Lastly, the fourth group of Gili Matra MPA stakeholders, who have low influence and interest, are known as the “Crowd” [21]. This group comprises the beneficiaries of the Gili Matra MPA in the form of a semi-formal organization. The stakeholders in this group are people who utilize the ecosystem services from Gili Matra as part of their livelihoods. As beneficiaries, this group of stakeholders does not have an essential contribution to the MPA management, and their interest in the MPA boundaries is limited to maximizing the ecosystem services. Their dynamics or changes in organizational circumstances are less likely to disturb the MPA. They also have a limited understanding of the MPA objectives and conservation efforts in Gili Matra. However, there could be some circumstances where these stakeholders could fall under other group categories, as they may still wait to see their influence and interest [32].

As outlined in this study, all of the interviewed stakeholders acknowledged the importance of the Gili Matra MPA’s influence over prosperity, with the primary source of the community wealth being marine tourism that functions to sustain the people’s future livelihoods. However, the stakeholders do not acknowledge the impact process of the MPA on prosperity in the same way. The fundamental difference is how the management approach is delivered to the people. The influential stakeholders think that the current state governing approach is the best way to sustain prosperity, while the NISs demand that the initiative and implementation to protect the ecosystem in the area should come from the communities. The difference highlights a vulnerability in delivering the MPA goals and its impact on the coastal communities, as it may indicate an inharmonious relationship between the stakeholders. The situation contradicts the ideal circumstances to achieve the best outcome of the MPA for the coastal communities, as the success of the MPA requires enduring commitment and engagement and reconciled communication among stakeholders to have shared values [33]. For example, the successful MPA in north-central California is due to transparency and trust among the stakeholders at multiple levels, accommodating local knowledge and public participation in the planning and implementation process [34]. In the worst scenario, the intended objectives of the MPA will potentially fail, as the management could lose support from the NISs if authorities do not consider their concerns. The NISs are comprised of academicians, NGOs, and public associations that play an essential role in supporting the conservation works in the MPA. These institutions could provide the MPA with a wide range of assistance, including funding, research, improving public awareness, monitoring, and enforcement [35,36,37]. Therefore, a failure to establish strong partnerships with the NISs may negatively affect the MPA management.

The state government is known for its top-down approach, while the community wants to establish a bottom-up approach. In MPA governance, the top-down approaches are translated through law and regulation by the state; meanwhile, bottom-up approaches are derived from participatory community design [35]. Both methods should be applied and substituted in MPA management despite these differences. Successful MPA planning will depend on balancing these approaches and how they are implemented [38]. The balance between top-down and bottom-up approaches in MPA management could lead to resource extractors and environmental activists reaching a similar perspective concerning MPA governance [39]. The combination of top-down and bottom-up approaches from MPA management will have arrangement variability between the community and state and will be designed according to the local context in socio-ecological aspects [40]. However, the mental model shows some disturbance of trust among the communities (represented by NISs) regarding the current MPA authority, as they believe the MPA should be locally managed. This situation must be improved if Gili Matra decides to apply the top-down and bottom-up approaches, considering that the mentioned scheme in the MPA requires people’s trust as the primary foundation [40]. Further, the balanced approach between local participation and government regulation will be beneficial to establishing MPA networks, which could provide broader impacts for marine conservation and expand the prosperity influence in other areas [41].

To avoid the scenario mentioned above, an approach to enhancing and adjusting the co-management structure within MPA development should be considered. Co-management in an MPA is known to improve the effectiveness of the MPA in achieving the intended conservation objectives, including optimizing the prosperity in communities. It can mitigate irresponsible management and sustain the existence of natural resources in MPAs [42]. One of the essential functions of implementing co-management in MPA governance is to reduce the potency of conflict among the stakeholders; thus, the local communities could achieve optimal prosperity [43]. Further, co-management applications are expected to introduce strong ecotourism to the MPA [44].

The situation shown in this report indicates that the governance and decision making of the MPA in Gili Matra are centered around the national government representatives, while community involvement in MPA governance is limited to public consultation and efforts to accommodate people’s aspirations. Applying co-management in Gili Matra should shift the power dynamic and require the government representatives to share their control with other groups [44]. Further, the shared influence in the MPA should be passed to the agreed-upon community representatives. Local community involvement in the governance and decision making is the key to enhanced co-management in the MPA [42,45].

From the perspective of the influential stakeholders, co-management will result in power shifting. By applying this scheme, the NISs may have more power and authority; thus, they could become influential stakeholders. The converted power on low-influence stakeholders could accelerate the sustaining process of the MPA impact by distributing administered skills among the community-level stakeholders [46].

Another potential co-management adaptation is through cultural practices in conservation. Gili Matra is known for its traditional approach to managing marine resources called *awig-awig* [47]. *Awig-awig* provides local wisdom power to the community to control a wide range of local governance, from the taxation of local ventures to urban planning, safety, and security, including marine resources utilization management and stewardship [48]. The implementation of *awig-awig* also involves sanctions from the local community for multi-level societies [48]. Strengthening the acknowledgment and inclusion of *awig-awig* could be one way to apply co-management between the government and community governance for the MPA in Gili Matra. At the current stage, *awig-awig* may be included as an NIS, as the co-management in the Gili Matra MPA is still limited to the MPA program participation [48]. In applying co-management enhancement, the *awig-awig* should be shifted into influential stakeholders by sharing the scheme’s decision-making roles. The MPA should then achieve the intended objectives as a supporting scheme among stakeholders; thus, the intended prosperity impact in a wide range of levels can be reached.

## 5. Conclusions

Through in-depth interviews and mental model development, the study found a difference between the influential and non-influential stakeholders of the MPA in acknowledging the impact of MPA management on local prosperity. Both categories of stakeholders understand that the existence of the MPA in Gili Matra is functioning to sustain the prosperity sourced from the area’s massive tourism. They think that the preserved ocean, as the outcome of the MPA approach, will provide them with ecosystem services and stabilize the tourism dynamic in the area; thus, the communities could afford all the requirements to live prosperously. However, the non-influential stakeholders of the MPA argued that the current MPA management, which is controlled mainly by the influential stakeholders, is ineffective in achieving the MPA’s social and economic outcome. They demand that the MPA approach maximize the local approach to be the dominant management direction. From the mentioned findings, it is beneficial for the MPA management to adjust the management approach to ensure support from the majority of the stakeholders. The stakeholders’ positions could be re-evaluated to see how the effectiveness of governance influences people’s wealth from MPA management.

This research is limited to the qualitative approach, and the findings are still in their preliminary phase. To reach a better understanding, a long-term quantitative assessment exposing the temporal change or spatial comparison of prosperity in the Gili Matra MPA needs to be conducted.

## Figures and Tables

**Figure 1 ijerph-19-13508-f001:**
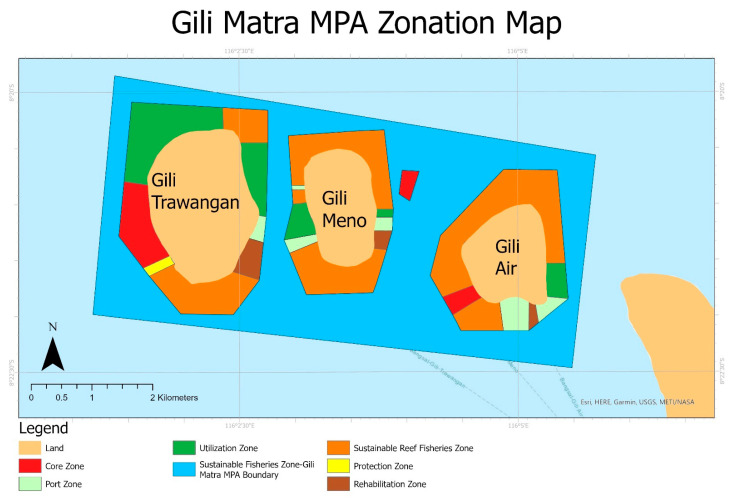
The zonation map of the Gili Matra MPA. The map is produced by using the MPA zonation information from the Ministry of Marine Affairs and Fisheries (MMAF), Indonesia [15] which used as the MPA management planning guidance.

**Figure 2 ijerph-19-13508-f002:**
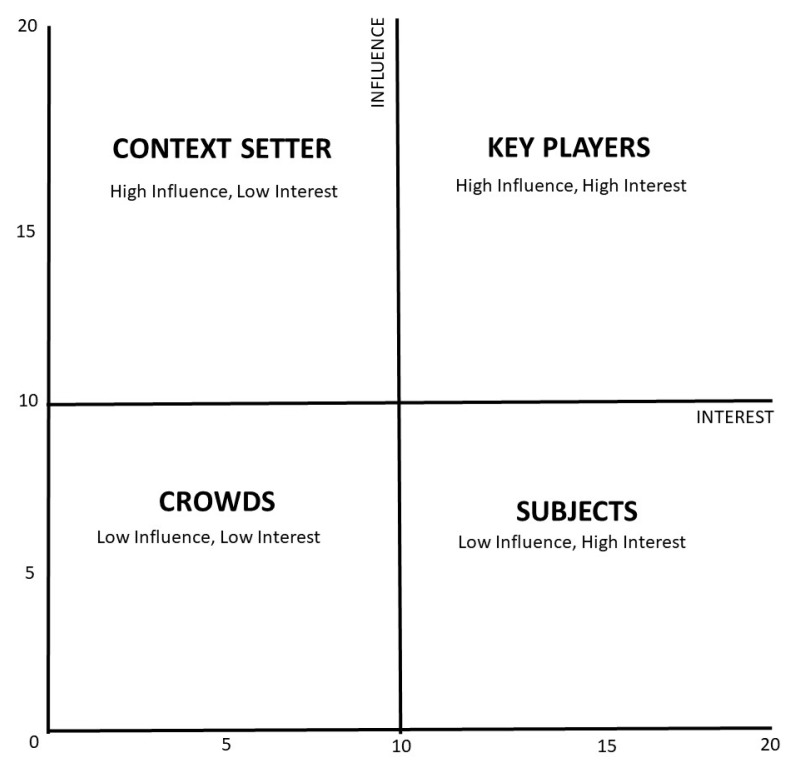
Adapted and modified grid [21,22] for stakeholder identification, which divides the stakeholder groups based on their influence and interest.

**Figure 3 ijerph-19-13508-f003:**
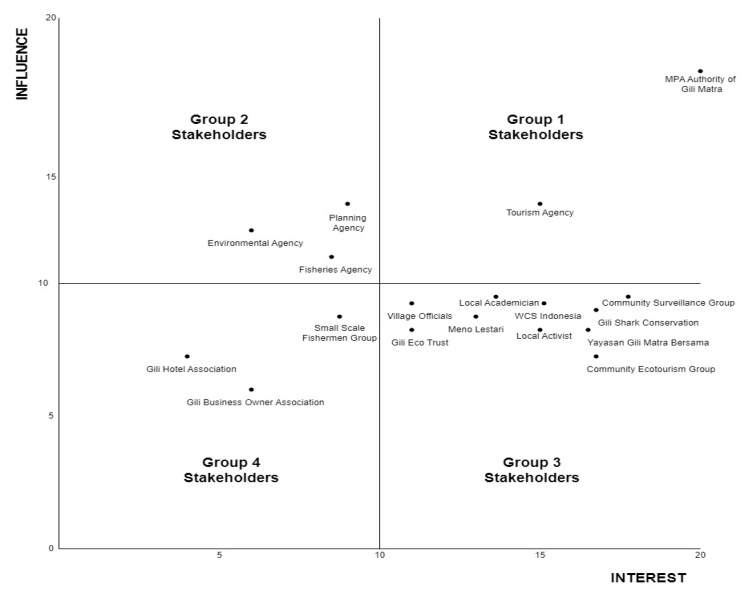
Matrix of stakeholder mapping results in the Gili Matra MPA. Out of eighteen stakeholders, only five of them are considered to have a significant influence score and possess authoritative work in the MPA, while the rest are considered as non-influential stakeholders.

**Figure 4 ijerph-19-13508-f004:**
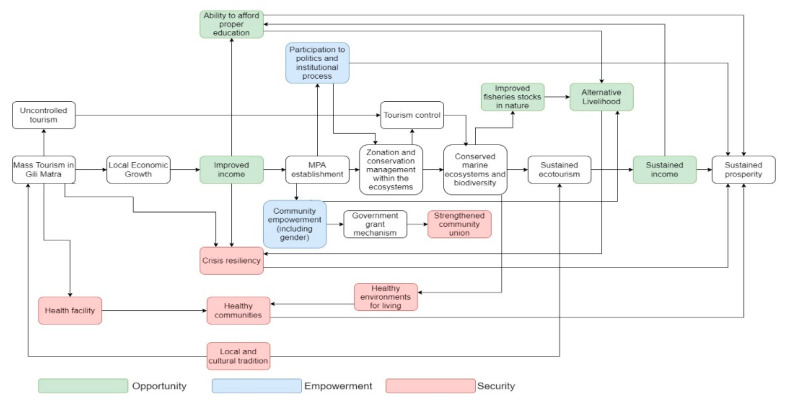
Mental model of influential stakeholders in the Gili Matra MPA and the MPA’s influence on local prosperity. Each box represents a situation in Gili Matra that is related to the prosperity and MPA management in Gili Matra. The colored boxes indicate their categorization to the World Bank domains of prosperity (opportunity, empowerment, security), while the transparent boxes indicate no inclusion regarding the mentioned domains. The mentioned stakeholders emphasize the importance of the MPA in providing sustained future prosperity.

**Figure 5 ijerph-19-13508-f005:**
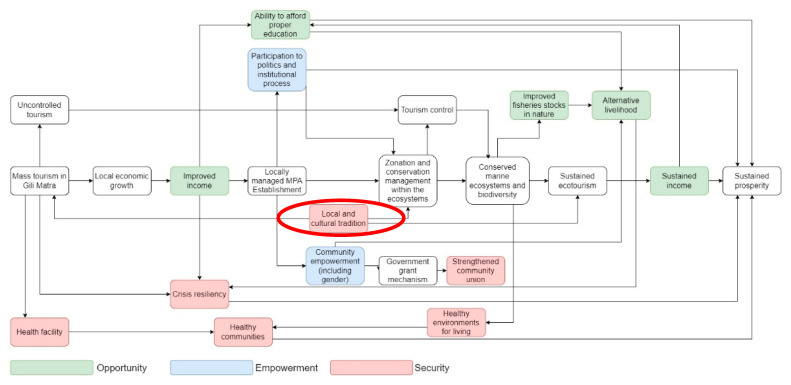
Mental model of non-influential stakeholders in the Gili Matra MPA regarding the MPA’s influence on prosperity. Each box represents a situation in Gili Matra that is related to the prosperity and MPA management in Gili Matra. The colored boxes indicate their categorization to the World Bank domains of prosperity (opportunity, empowerment, security), while the transparent boxes indicate no inclusion with the mentioned domains. The aforementioned stakeholders have a similar way of thinking compared to the influential stakeholders, with the exception of the management delivery, where the local management approach is believed to have a greater impact (red oval).

**Table 1 ijerph-19-13508-t001:** Adapted scoring indicators [23,24] for stakeholder mapping tools in the Gili Matra MPA. Each indicator will provide 1–5 scores; thus, the total score for each axis in the grid has a maximum score of 20.

Influence Indicators	Interest Indicators
Involvement in Policy Establishment	Involvement in Management
Resource Capacity	Achieved Benefit
Jurisdiction to Manage	Activities
Contribution and Participation	Dependence

**Table 2 ijerph-19-13508-t002:** Description of each respondent stakeholder in the Gili Matra MPA.

Name of Stakeholder	Description
Influential Stakeholders (IS)	
MPA Authority of Gili Matra	This is the primary managing institution of the MPA in Gili Matra. They were appointed by the National Government of Indonesia through the Ministry of Marine Affairs and Fisheries
Tourism Agency	Responsible for managing tourism activity in the district, including improving the tourism experience quality in the Gili Matra to attract more visitors.
Planning Agency	Designing the district plan and monitoring its implementation through policy approaches, including designing a district revenue stream from the Gili Matra MPA.
Fisheries Agency	Managing the fishing activity of the local district, including the fisheries activity in the Gili Matra MPA
Environmental Agency	Managing the environmental issue of the local district, including the waste management in the Gili Matra MPA
Non-Influential Stakeholders (NIS)	
Community Surveillance Group	Local community groups voluntarily surveilling the marine area of the Gili Matra MPA for violations of the MPA zonation and other illegal marine extraction activities
Gili Shark Conservation	Local NGO focusing on shark conservation in the Gili Matra MPA
Yayasan Gili Matra Bersama	Local NGO focusing on marine conservation and sustainable tourism in the Gili Matra MPA
Community Ecotourism Group	Local community group focusing on spreading awareness and practicing community-based sustainable tourism in the Gili Matra MPA
Local Activist	Representation of a group of people who actively spread awareness of sustainable tourism and marine conservation in the Gili Matra MPA
WCS Indonesia	National NGO working on marine conservation. The Gili Matra MPA is one of their site project locations
Local Academician	Representation of local researchers and academics who actively conduct a wide range of research and study programs in the Gili Matra MPA
Meno Lestari	Local NGO focusing on marine conservation and sustainable tourism in the Gili Matra MPA
Village Officials	Government officials responsible for managing the civil administration of the people in the Gili Matra MPA
Gili Eco Trust	Local NGO focusing on marine conservation and sustainable tourism in the Gili Matra MPA
Gili Hotel Association	Association for hotel managers in the Gili Matra MPA
Gili Business Owner Association	Association for tourism venture owners in Gili Matra. Their members comprise hotels, restaurants, trip operators, caterers, and boat and bar owners in the Gili Matra MPA
Small Scale Fishermen Group	Local community organization specifically for the fishermen in the Gili Matra MPA

## Data Availability

Not applicable.

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
