# Peer review of "Understanding How Marine Protected Areas Influence Local Prosperity—A Case Study of Gili Matra, Indonesia"

_ijerph, 2022, doi:10.3390/ijerph192013508_

Round 1

Reviewer 1 Report

The topic is interesting, but should be better explained, apart from the differentiation between IS and NIS, because both groups are themselves divided into two subgroups, if the interviews are then only classified as IS and NIS. 

The paper would also need a boost on the quantitative side. The number of interviews conducted and the relationships between IS and NIS should be highlighted.

Figure 4 lacks captionn due to editing issues. Why are there three colours with their respective captions in figures 3 and 4 and some boxes are simply transparent?

Reviewer 2 Report

This article is clear and informative, several comments and suggestions are:

1.     Please provide which year these interviews were conducted in the article.

2.     Is “eighteen stakeholders were identified (Line 121)” means there were 18 people being interviewed or more people were interviewed? Please clarify.

3.     In Figure 1, the colors used to represent the “Sustainable Reef Fisheries Sub Zone” and the “Utilization Zone” cannot be distinguished. Are they the same? If not, please revise.

4.     There were too many interpretations in the “Results” sections, they should be considered to move to Discussion. ex. Line 162-165, 175-177, 214-216, 225-227.

5.     The mapping results showed in Figure 2 require a description of the methodology in the “Materials and Methods” section.

6.     Please provide supporting data for statements of “Their duties are less likely to affect the MPA management, but their interest in developing the area’s community supports the MPA works indirectly (Line 212-214)” and “As beneficiaries, this group of stakeholders does not hold an essential contribution to the MPA management, and their interest in the MPA boundaries is limited to maximizing the ecosystem services. Their dynamics or changes in organizational circumstances are less likely to disturb the MPA. They also have a limited understanding of the MPA objectives and conservation efforts in Gili Matra (Lin 221-225).” Otherwise maybe is better to move them to the “Discussion” section.

7.     Consider to move the sentences in Line 236-241 (“This report will explore…..”) to “Materials and Methods” section.

8.     No supporting data was provided for the sentences of “Besides this resiliency, the security dimension is also affected by the strong sense of social organization (Line 186-187)”. Please provide or modify.

9.     Consider to highlight those elements in Figure 4 which were different from Figure 3, so readers can pick up the differences quickly. Also, Figure 4 covered sentences in Line 477-479, needs to be corrected.

10.  In Conclusions, consider to use “difference” to replace “distinct difference” in Line 560-561.

Reviewer 3 Report

In general, this research is well written and set up a model well. However, since this study is qualitative research, needs to show how this research is processed scientifically. In addition, needs to provide demographic information about the interviewee.

Regarding the stakeholder group, I recommend you make a table and show the information.
